



# A high transmission axial ion mobility classifier for mass-mobility measurements of atmospheric ions

Markus Leiminger[1,3], Lukas Christoph Fischer[1], Sophia Brilke[2,*], Julian Resch[2,#], Paul M. Winkler[2], Armin Hansel[1] and Gerhard Steiner[1,4]

[1]University of Innsbruck, Institute of Ion Physics and Applied Physics, 6020 Innsbruck, Austria
[2]University of Vienna, Faculty of Physics, 1090, Vienna, Austria
[3]Ionicon Analytik GmbH, 6020 Innsbruck, Austria
[4]Grimm Aerosol Technik Ainring GmbH & Co. KG, 83404 Ainring, Germany
[#]now at: University of Basel, Department of Environmental Sciences, 4056 Basel, Switzerland
[*]now at: IMS Nanofabrication GmbH, Austria

*Correspondence to*: G. Steiner (gerhard.steiner@grimm.durag.com) and A. Hansel (armin.hansel@uibk.ac.at)

**Abstract.** We present an electrical mobility classifier for mass-mobility measurements of atmospheric ions. Size segregation coupled with mass spectrometric detection of naturally occurring ions in the atmosphere is challenging due to the low ion concentration. Conventional electrical mobility classifying devices were not yet coupled with mass spectrometry to resolve natural ion composition. This is either due to the insufficient transmission efficiency, or design concepts are incompatible with this application e.g., using high electric fields close to the inlets to push ions from high to low electric potential. Here, we introduce an axial ion mobility classifier, termed AMC, with the aim to achieve higher transmission efficiencies to segregate natural ions at reasonable sizing resolution. Similar, to the recently introduced principle of the high-pass electrical mobility filter (HP-EMF) presented by Bezantakos et al., 2015, and Surawski et al., 2017, ions are classified via an electric field that is opposed to the gas flow direction carrying the ions. Compared to the HP-EMF concept, we make use of sheath flows to improve the size resolution in the sub 3 nm range. With our new design we achieve a sizing resolution of 7 with a transmission efficiency of about 70 %.

## 1 Introduction

Ions are omnipresent in the Earth's atmosphere due to the production from galactic cosmic rays and radioactive decay of primarily Radon (Rn). The noble gas Rn is a radioactive intermediate from the thorium and uranium decay chain. Since uranium-containing minerals in soil vary from place-to-place Rn levels differ. The number concentration of ions range from several hundred to a few thousand ions per cm³ in the atmosphere (Hirsikko et al., 2011). First mass spectrometric studies of tropospheric ions were done by Eisele and Perkins in the 1980's (Perkins and Eisele, 1984). Due to the limited mass resolving power early mass spectrometric investigations of atmospheric ions made use of collision induced dissociation (CID) methods to identify the core ions of ambient clusters like $NH_4^+$ or protonated amines e.g. pyridine (Eisele, 1988). With the introduction of the atmospheric pressure interface time-of-flight (APi-TOF) mass spectrometer, high mass resolving power allowed direct attribution of sum formulas even to molecular clusters and further extended the detectable mass range enabling the observation of clusters, neutral and naturally charged ones (Junninen et al., 2010; Kürten et al., 2014; Kirkby et al., 2011).

The APi-TOF and other new instrumentation brought new insights in the field of new particle formation (NPF), which is about the formation of molecular clusters and growth into nanoparticles from vapours (Kulmala et al., 2001). Ions were found to be able to contribute to NPF via ion-induced nucleation (IIN) which is energetically favoured over the neutral pathway (Curtius et al., 2006; Kirkby et al., 2016). NPF including the neutral and ion-indued pathway is estimated to contribute globally to more than half of the



total particles number concentration and is therefore important for our climate system regarding the radiative forcing of aerosols
(Boucher et al., 2013; Dunne et al., 2016).

Motivated from frequently observed NPF events in the atmosphere numerous experiments have been conducted involving ions and neutral particles in the CLOUD chamber at CERN during recent years (Simon et al., 2020; Almeida et al., 2013; Kirkby et al., 2011; Schobesberger et al., 2013; He et al., 2021). NPF in the remote atmosphere involves extremely low volatility compounds such as sulfuric acid or highly-oxygenated organic molecules (HOM), and vapours such as ammonia or amines having the ability

to stabilize freshly formed clusters (Tröstl et al., 2016; Kirkby et al., 2011; Bianchi et al., 2016). During wintertime (below +5 degrees Celsius) NPF in urban areas can occur when nitric acid and ammonia vapours condense on freshly nucleated nanoparticles achieving growth rates fast enough to compete with scavenging losses (Wang et al., 2020). IIN could become the dominant process at very low condensing vapour concentrations. In pre-industrial times with no anthropogenic sulphur emissions sulfuric acid concentrations were rather low. Under these conditions HOMs, which are produced through photooxidation of biogenically emitted

VOC precursor gases can produce aerosol particles. Galactic cosmic rays increase the nucleation rate by one to two orders of magnitude (Kirkby et al., 2016). Recently He et al., 2021, showed that the nucleation rate of iodine oxoacids exceeds that of the $H_2SO_4$-$NH_3$ system at the same acid concentrations. Global iodine emissions have increased threefold over the past 70 years and may continue to increase in the future. Iodic acid ($HIO_3$) is the major iodine species driving both nucleation and growth in pristine regions of the atmosphere such as marine coasts, the Arctic boundary layer, or the upper free troposphere. At low temperatures of

-10 degrees Celsius neutral nucleation is already fast and IIN does not contribute. In contrast, at temperatures of +10 degrees Celsius a strong IIN contribution at low iodic acid concentrations was observed.

In these studies, the atmospheric pressure interface time-of-flight (APi-TOF) mass spectrometry technique proved to be a valuable tool for the study of the chemical composition of the nucleating clusters and the identification of the involved species (Junninen et al. 2010). However, the transfer of molecular cluster ions through the different pressure stages of a mass spectrometer from ambient

pressure to the vacuum of a mass analyser can be affected by fragmentation induced by elevated electric fields of the ion transfer optics (Olenius et al., 2013; Ehrhart et al., 2016). In laboratory experiments the vapours are well-known allowing, in principle, the simulation of cluster distributions that can be compared with mass spectrometric data to identify and correct for fragmentation (Alfaouri et al., 2021). This is not straightforward in a natural environment. Therefore, the final goal of this study is the coupling of a mobility analyser with an APi-TOF as this is expected to facilitate direct identification of fragments during non-targeted online

measurements of a complex gas matrix.

Coupling a mobility filter like a differential mobility analyser (DMA) to the inlet of a mass spectrometer to reveal the decomposition of cluster ions has been used by several groups (Hogan and Fernández de la Mora 2009; Ouyang et al. 2015; Passananti et al. 2019). With the present study, we aim to use such an approach for studying atmospheric ions. But there is one obstacle: the low number concentration of ions in the atmosphere which ranges from several hundred to a few thousand ions per

cm³ in the atmosphere (Hirsikko et al., 2011). The transmission through an APi-TOF is below a few percent. The transmission through typical DMAs is comparably higher but typically below 50 % and, in most cases, even below 10 %. Therefore, coupling a DMA to an APi-TOF will result in rather low overall transmission efficiencies. There are DMAs that have in principle ideal characteristics like the p5 DMA of SEADM for example, a planar DMA, offering high resolution (>100) and high transmission (>50 %) (Amo-González and Pérez, 2018). However, the high voltage electrode is typically located at the inlet and the grounded

electrode on the opposite site of the entrance of the mass spectrometer. For the sampling of atmospheric ions, this would require moving them from atmospheric electric potential i.e., ground, through the high voltage of the DMA prior to classification. This would most likely result in increased losses.





Other DMAs, like the UDMA, half-mini DMA and cyDMA are designed for small electrical mobility diameters and offer suitable resolutions of 10 to 20 while their transmission efficiencies are not exceeding 12 % in best cases (Cai et al., 2017; Steiner et al., 80    2010; Wang et al., 2014).

Bezantakos et al. 2015, recently developed an axial electrical mobility filter using tubing made of electrostatic dissipative materials (EDM) for the size-segregation of nanoparticles in the size range from 10 to 55 nm where the sample is introduced against an opposed electric field (Bezantakos et al., 2015; Zeleny, 1929). The authors improved their sizing range down to 1 nm in Surawski et al., 2017, with the High-Pass Electrical Mobility Filter (HP-EMF). In contrast to a DMA which classifies ions as a band-pass 85    filter, the HP-EMF filters ions of higher mobility first while ions of lower mobility may still pass i.e., working as a high-pass filter. The simple and cost-effective idea was picked up to improve the DMA outlet problem in the recent literature and proved to increase transmission of DMAs by up to 50 % while not being used for size segregation (Attoui and de la Mora, 2016; Cai et al., 2018). Therefore, the working principle of the HP-EMF offers promising characteristics regarding its transmission besides its currently rather low sizing resolution.

Here, we present a revised approach of the axial size segregation principle that we introduce as the axial mobility classifier (AMC). An axial electric field is directed against the direction of the aerosol flow like in the HP-EMF. In contrast to previous designs presented in the literature, we use ring electrodes made of stainless steel instead of EDM materials and we incorporate a sheath flow to avoid radial velocity gradients within the sample gas flow to improve the sizing resolution. Using artificially created ions we characterize the transmission and the sizing resolution of the AMC for a range of flow conditions. Finally, we present first 95    proof-of-principle mass-mobility measurements of atmospheric ions by combining the AMC with an APi-TOF.

## 2 Methods

### 2.1. The Axial ion Mobility Classifier (AMC)

### 2.1.1 Working principle

The Axial ion Mobility Classifier (AMC) uses the principle of segregating ions with a flow opposed to the electric field as 100    illustrated in Fig. 1, which follows the idea of Bezantakos et al., 2015, and which was also proposed for an inverted drift tube (Nahin et al., 2017). Here, ions move in the direction of the aerosol flow at a velocity $\vec{v}$ while the electric field $\vec{E}$ is directed in the opposite direction. In a simplified picture, the electrical mobility Z is related to the electric force and the drift velocity in the electric field.

$$\vec{v} = Z \cdot \vec{E} \tag{1}$$

The axial electric field decelerates the ions, which are subsequently pushed to the walls via diffusion and the radial component of 105    the electric field and finally removed from the aerosol flow. The combination of flow rate and classification voltage determines a general critical electrical mobility $Z^*$ threshold. Only ions that have a lower electrical mobility Z than this critical mobility can pass the potential barrier $\Phi$ while ions with a higher electrical mobility cannot pass. Following Suraswki et al., 2017, the relative penetration efficiency $P_r(\Phi)$ through the device is determined as:

$$P_r(\Phi) = \begin{cases} 1 - \dfrac{Z}{Z^*}, & Z < Z^* \\ 0, & Z \geq Z^* \end{cases} \tag{2}$$

We term the electrical mobility of a specific ion where 50 % of the initial number concentration can be detected as the half-pass mobility $Z_{1/2}$.



### 2.1.2 Design concept

All conductive parts of the device are made of stainless-steel; insulators were made from Teflon. Ions enter the inlet in axial direction through a ½" stainless-steel tube as shown in Fig. 2. For aerosol flow rates of up to 15 L/min, flow conditions are laminar.

These dimensions were preferred for the design of the inlet tube to reduce wall losses due to the low abundance of atmospheric ions. Within the device, the inlet tube is centred in a tube of a larger diameter. The outer tube has four inlet ports where sheath air is introduced with an angle of 90° to the centre axis. A screen made of nylon with a mesh size of 50 µm is stretched between the inner and outer tube to laminarize the sheath flow.

Downstream of the screen, we designed a cone with a length of 65 mm forcing the flow to accelerate as the inner tube diameter is

reduced from 26 to 10 mm (Perez-Lorenzo et al., 2020). To reduce turbulence formation, the cone angle is kept small with 7° inclination. Within the cone, the aerosol flow meets the sheath flow. The diameter of the tube after the cone has 10 mm which is the same as the inlet tube. The main classification region is between the end of the cone and the labelled electrode, the AMC lens. The AMC can technically be run either in a voltage-scanning mode (AMC scan) where the voltage is iteratively changed from e.g., low to high or in a constant voltage mode at a fixed voltage. In this study, we only present data of the scanning mode.

Teflon insulators having the same inner diameter of 10 mm separate the lens to the neighbouring metal parts. The Teflon insulator between the cone and the AMC lens has a width of 4 mm to allow the application of voltages up to 8 kV at atmospheric pressure. The width of the Teflon insulator downstream of the AMC lens is 20 mm as it is not critical for the ion classification but reduces the electric field strength in the post-classification region.

After passing the classification region, size segregated ions are collected via a core sampling (Fu et al. 2019). As recommended by

Fu et al., 2019, the loss parameter is below 0.1 for all flow rates larger than 2 L/min. The core sampling is connected to a detector either a Faraday Cup Electrometer (FCE) or a mass spectrometer that transfers the ions at a flow rate of 1 L/min. The rest of the aerosol is pumped radially to the outlet ports, together with the sheath flow. The sheath flow is recirculated via a brushless blower (model code: 465.3.265-841) from Domel, Slovenia, at experimentally determined flow rates of approximately 50 to 115 L/min.

### 2.2. Experiments for instrument characterization

A bipolar electrospray ionization source (ESI, Fernandez de la Mora and Barrios-Collado 2017) was used for ion generation. The benefit of using a bipolar ESI is that charged droplets of opposite polarity can be generated leading to a favourable reduction of multiple charged ions enabling the precise formation of cluster ions of 1 to 4 nm as previously shown in (Brilke et al., 2020).

We characterized the transmission through the AMC with the experimental setup shown in Fig. 3 A. The setup consists of the bipolar ESI, the Vienna-type UDMA (Steiner et al., 2010), two Faraday Cup Electrometers (FCE, Winklmayr et al. 1991) and the

AMC. Cleaned and dried air at flow rates between 10 and up to 24 L/min carried the ions into the UDMA. The UDMA is used in scanning mode for acquiring ion mobility spectra and in constant voltage mode for selecting a monodisperse stream of a cluster ion. A y-shaped flow splitter (inner diameter 8 mm) with a tube splitting angle of 60° is placed behind the UDMA. One flow is directed to an FCE (FCE1) while the other is transferred through the AMC that is mounted directly in front of the second FCE (FCE2).

In our experiments, we used millimolar solutions of Tetraheptylammonium bromide (THAB) and Tetrapropylammonium iodide (TPPAI) solved in acetonitrile (Ude and De La Mora, 2005). With the UDMA, we selected well-defined cluster ions e.g., from the monomer up to the pentamer in case of THAB corresponding to a mass-to-charge (m/z) ratio of 410.47 Th up to 2368.03 Th or in terms of inverse mobility a range from 0.619 to 2.486 Vs/cm².

We characterized the AMC at three different aerosol flow rates, $Q_{ae}$: 5.0, 9.6 and 12.4 L/min, respectively. For each aerosol flow

setting, the same flow, $Q_{ae}$, was also introduced into FCE1. While FCE1 detects ions at the flow rate of $Q_{ae}$, the flow through the





core sampling into FCE2 is always set to 1 L/min to simulate the flow rate into the mass spectrometer in the following experiments. Ion count rates are corrected on these flow differences. The transmission efficiency of the AMC was further calibrated for five different blower settings corresponding to different sheath flow rates. The blower was set to voltages from 0 to 5 V in 1 V steps.

### 2.3. Mass spectrometer

The ioniAPi-TOF mass spectrometer, Ionicon Analytik GmbH, Austria, was recently introduced by Leiminger et al., 2019. It is made of an Atmospheric-Pressure interface (APi) that consists of two hexapole ion guides as well as an ion transfer optic and an orthogonal extraction, reflectron Time-Of-Flight (TOF) mass analyser. During these experiments, we used a compact ioniAPi-TOF 1000 and an ioniAPi-TOF 6000 with mass resolutions tuned to 1700 and 5500 for ions larger than 100 m/z. The compact ioniAPi-TOF was used to validate the chemical composition of the cluster ions used in the previous experiments. The higher

resolving ioniAPi-TOF was deployed for exemplary proof-of-principle mass-mobility measurements of atmospheric ions with the combined AMC-ioniAPi-TOF approach. Here, a ½" stainless-steel tube of approximately 0.5 m length was connected to the inlet of the AMC with the other end standing out through the window with a length of 0.2 m.

We used the Julia language-based TOF-Tracer2 data processing scripts (https://github.com/lukasfischer83/TOFTracer2, last access: 15 December 2021; https://doi.org/10.5281/zenodo.5781695, Fischer, L., Leiminger, M., and Eccli, E; 2021) and adopted

them for data analysis of the ioniAPi-TOF. High-resolved peaks were fitted and sum formulas attributed with the software peakFit (https://github.com/lukasfischer83/peakFit, last access: 15 December 2021; https://doi.org/10.5281/zenodo.5781711, Fischer and Breitenlechner, 2021). For more details we refer the reader to Breitenlechner et al., 2017, and Fischer et al., 2021. The Ionicon PTR-MS Viewer 3.2 was used for data analysis of the compact ioniAPi-TOF.

In the application of the AMC-ioniAPi-TOF for measurement of atmospheric ions, a new measurement cycle method was

developed to counteract the limited abundance of the ions. The AMC voltage was scanned in 15 min from 0 to 1800 V with voltage steps of 12 V. This cycle was repeated throughout the measurements. The AMC scan was synchronized with the ioniAPi-TOF. Voltage steps with the AMC and mass spectra were saved at one second time resolution. The data analysis involved the following steps. Mass spectra were mass axis calibrated. For Fig. 9, we created time-resolved traces by integrating peaks of unit mass resolution (UMR) within -0.2 and +0.4 for each nominal m/z in the mass range of 17 to 900. All the time steps were sorted for the

AMC voltage. From 0 to 1800 V, UMR traces were binned and averaged in steps of 26 V.

### 2.4. Flow simulations with OpenFOAM

We used the OpenFOAM version 7 (https://openfoam.org/) in a Linux environment (Ubuntu 18.04) to simulate the flow in the AMC inlet. The geometry was designed in the software freeCAD version 0.18 (https://www.freecadweb.org/). The mesh and the simulation case were created in the software HelyxOS (https://engys.com/products/helyx-os). We selected the simpleFoam solver

in the incompressible flow approximation. Turbulence was accounted for with the k-ω-SST turbulence model. As an exemplary maximum flow rate, we estimated from the laboratory experiments to be about 105 L/min which corresponds to roughly 22.3 m/s for the classifier's tube diameter. This is below 8 % of the Mach speed. Therefore, an incompressible flow description is reasonable for all tested conditions. Typically, the transition to a compressible fluid description is recommended above 20 % of the Mach speed (Oertel, Böhle, & Dohrmann, 2009, p. 106). Additionally, an electric field solver was created for the visualization of the

field lines.





## 3. Results

### 3.1. CFD simulations

The flow and electric field in the AMC were simulated to gain better insights for designing the instrument and for interpretation of the experimental results. There are three main parts that require attention, the mixing region of sheath and aerosol flow, the
classification region, and the core sampling region.

In the classification region, it is important to offer homogeneous classification conditions by matching the electric field and the flow field. The laminar flow profile in the HP-EMF design pushing ions against the homogenous electric field result in suboptimal sizing resolution. Here, a plug flow would offer an idealized radially flat velocity profile of reduced axial velocity spreads. To realize a plug flow, the AMC uses a sheath flow like classic DMAs to improve the sizing resolution. The mixing of sheath flow
and aerosol flow must be well-designed such that mixing of both flows and consequently dilution of the sample does not appear. This method is also known from Eisele-type flow tube chemical ionization sources (Eisele and Tanner 1993). Additionally, turbulence from flow mixing must be avoided. The concept of coaxial mixing of an outer and an inner flow with high velocity ratios is well studied in the literature where it is known as coaxial jets and relevant for the mixing of fluids of different media e.g., in combustion chambers of rocket engines (Rehab et al., 1997). In such systems, the mixing of two fluids can lead to instabilities
at their shear layer like the Kelvin-Helmholtz instability. These instabilities can result in turbulent mixing of both fluids and are favoured by larger differences in density, temperature, or flow velocity. In our case, two coaxial gas flows were used having the same density and rather similar temperatures, but notably different velocities that could in principle induce turbulences in terms of sub-isokinetic mixing.

A colour plot of the fluid simulation of the AMC can be found in Fig. S7. The flow profiles for different sheath to aerosol flow
ratios are extracted from the presumed position of the mobility classification and are presented in the upper panel of Fig. 4. A typical Hagen-Poiseuille velocity profile develops if only the aerosol flow is introduced into the AMC while the sheath flow is off. With an additional sheath flow, the flow profile evolves almost into the desired plug flow profile for a combination of 10 L/min sample and 55 L/min sheath flow. With higher sheath flow rates, it results in an M-shaped velocity profile due to the fluid velocity being faster at larger radii than the velocity in the central axis. The highest total flow rate corresponds to a Reynolds number of
about 17000. At a medium sheath flow setting of 55 L/min the impact of aerosol flow rate becomes clear comparing the flow profiles for aerosol flows of 5 and 10 L/min.

For the sheath to aerosol flow ratio of 55/10, we investigated the flow profile in z-direction, shown in the lower panel of Fig. 4, and found that the main features like the mixing region of sheath and aerosol flow clearly demonstrated the acceleration in the tapered region as well as a sharp drop in flow velocity as the sample is extracted into the core sampling. Overall, the simulation
results demonstrate that the principal ideas e.g., flow acceleration using a cone and the plug flow scenario, could be realized under certain conditions.

### 3.2. Characterization of the AMC

### 3.2.1 Example AMC scan

The general working principle of the AMC is experimentally demonstrated in Figure 5Fig. 5 as an AMC scan using a constant
stream of THAB dimer ions with the setup in Fig. 3A. Ions were detected with an FCE. The relative penetration efficiency is quasi constant if the electrical mobility of the THAB dimer $Z_d$ is lower than the half-pass mobility $Z_{1/2}$. The signal of THAB dimer ions starts to decrease as the voltage approaches the half-pass mobility. Finally, no ions can pass the electric potential barrier because their electrical mobility is above the half-pass mobility ($Z_d > Z_{1/2}$).





To obtain information about the performance characteristics of the AMC, we used a sigmoid function to describe the behaviour of the signals. The derivative of the sigmoid function yields a pseudo-peak. From this pseudo-peak, we obtain the full width at half maximum (FWHM), $\Delta Z$. The half-pass mobility at the maximum of the pseudo-peak corresponds to the electrical mobility of the ion of interest. Finally, we derive the resolution of the AMC according to (Flagan, 1999) as:

$$R = \frac{Z_{1/2}}{\Delta Z} \tag{3}$$


For simplicity, we will use either half-pass mobility or half-pass voltage with the same meaning while the latter one will be used for scenarios where the mobility was not determined.

### 3.2.2 Role of aerosol and sheath flow

In the tandem UDMA-AMC-FCE experiment, see Fig. 3A, the focus was characterizing the AMC at different flow rate settings
for aerosol and sheath-flow, respectively. Fig. 6 shows 20 individual AMC scans for two selected mobility standards, THAB monomer and dimer. More data for other mobility standards can be found in Fig. S1. As can be seen in all four panels in Fig. 6, the higher the sheath-flow, the higher the required voltage to reduce the fraction of passing ions and to finally filter all ions. Comparing panels, A and B, the size segregation between THAB monomer and dimer ions becomes clear. From panel B to C to D, the half-pass voltage gets lower in line with the aerosol flow rates that are 12.4, 9.6 and 5.0 L/min, respectively.
In the case of the THAB monomer, we observed that the lower end of the slopes does not decline as smoothly to zero compared to the other ions as they approach approximately 10 to 20 % of their initial intensity. Instead, they show a second slope which becomes more pronounced at higher sheath flows in line with a higher resolution. The same was also observed for the TPPAI monomer, see Fig. S1. The second slope results most likely from an impurity compound in the setup that has a lower electrical mobility compared to the THAB monomer itself. Thus, it is classified at a higher half-pass voltage. We confirmed this hypothesis from measurements
made using a tandem UDMA and a tandem UDMA-AMC-ioniAPi-TOF setup, see Fig. S2, S3 and S4. We found that the second slope is a cluster at m/z 800 of the THAB monomer with a yet unidentified compound. Similar experiments were not performed for the TPPAI monomer.

We also considered a space charge effect on the surface of the Teflon insulators between the electrodes as this was discussed in Franchin et al., 2016, for a different setup. The authors attributed this effect to the use of unipolar ions that were utilized for
instrument characterization. This effect cannot be ruled out completely, but it seems to be less likely regarding the previous discussion. However, we expect to have a minor impact on the measurements of atmospheric ions due to their bipolar appearance as well as their dramatically lower abundance. To finally rule out space charges on surfaces in future attempts, this could be overcome using EDM tubes for example. Still, space charge induced fields from unipolar classification could detrimentally affect the sizing resolution (Higuera and Fernandez de la Mora, 2020).
In some of the experiments for a lower aerosol flow of 5 L/min, but also some for an aerosol flow of 9.6 L/min, we observed that the signal increased above the initial intensity scanning from low to high voltages before starting to decrease due to size segregation, see Fig. 5 panels C and D. As we compared all these experiments, we found that an increase in intensity can only be observed for special flow conditions, namely a high sheath flow to aerosol flow ($Q_{sh}/Q_{ae}$) ratio. We attributed the reason for this initial increase in transmission to the combined effect of sub-isokinetic sampling in the region of the core sampling and elevated electric fields
that might reach into the core sampling as is exemplarily illustrated in Fig. S5.





### 3.2.3 Characterization of transmission and resolution

The transmission through the AMC is determined by the ratio of the ion number concentration N from the two FCEs running in parallel i.e., $N_2/N_1$. The number concentration was calculated from the electric currents measured with the FCEs considering the different flow rates. FCE1 serves as a reference while FCE2 accounts for all ions that move through the AMC including losses. If no voltage is applied, all ions can pass the AMC. We corrected the number concentration for the different aerosol flow rates. We also switched the position of both FCEs and found a systematic difference of 1.09 between FCE2 and FCE1. We used this factor to correct all data from FCE2. The corrected ratio of $N_{FCE2(U=0V)}$ to $N_{FCE1}$ yields the transmission efficiency for a single setting.

$$\text{transmission} = \frac{N_{FCE2(U=0V)}}{N_{FCE1}} \tag{4}$$

The transmission efficiencies for the tested conditions are shown in Fig. 7. Panel A shows the results for 5.0 L/min, panel B corresponds to 9.6 L/min and panel C to 12.4 L/min. The transmission is given in relation to the ratio of the total flow through the AMC to the aerosol flow: $Q_{total}/Q_{ae}$. The total flow rate $Q_{total}$ is the sum of $Q_{ae}$ and the sheath flow $Q_{sh}$. $Q_{total}$ was calculated using the following simplified equation for cases where $Q_{sh}$ is larger than zero.

$$Q_{total} = \frac{\frac{Z_1}{2} \cdot \Delta\phi}{L_{eff}} \pi r^2 \tag{5}$$

Here, r is the tube radius, $Q_{total}$ the flow rate, $\Delta\phi$ the electric potential difference. The effective length $L_{eff}$ is the distance between the two classifying electrodes, which is typically 0.4 cm. We used the known electrical mobility of the respective mobility standard tetraalkylammonium ions as the half-pass mobility to determine the flow rates at different sheath flow settings. The determined flow rates are only valid close to the tube centre and neglect differences in the flow profile. Like Surawski et al., we needed to correct the value of the voltage that we applied to the classifying electrode because of the short distance between the classifying and the grounded electrode upstream. Using the simulated electric field in OpenFOAM, we determined a voltage at the centre of the AMC tube of about 568 V for 1000 V applied to the electrode. This is slightly lower compared to the 581 V in Surawski et al., 2017, for the EDM tubes in the HP-EMF.

In general, the transmission efficiency shows highest values of up to 70 % if no voltage is applied. For comparing the performance characteristics of the AMC with the previously mentioned DMAs, the given values were determined for THAB monomer at 1.48 nm at an aerosol to sheath flow ratio of approximately 1/10. To allow a fair comparison, the transmission at the half-pass mobility of the AMC should be considered which, in line with the definition of the half-pass mobility, means that the transmission efficiency of the AMC should be divided by 2. In this case, the AMC offers a transmission of 48.6 % when no voltage is applied. This value translates in a DMA-equivalent transmission of 24.3 % that is higher compared to the performance of other DMAs like the half-mini DMA with a transmission of roughly 12 %.

The results of the sizing resolution in relation to the ratio $Q_{total}/Q_{ae}$ are shown in the lower panels of Fig. 7. The uncertainties are higher in determining the sizing resolution for cases where the transmission efficiency was low as this resulted in higher signal-to-noise. Consequently, this is mainly the case for experiments without sheath flow. Further uncertainties in the determination of the resolution result from the increased transmission that was discussed in the previous section and that mainly affects the experiments with the low aerosol flow rate of 5 L/min as well as the measurements at an aerosol flow of 9.6 L/min in combination with the maximum sheath flow setting. To obtain meaningful values for the resolution, the data points of the increasing values were ignored





for the sigmoidal fit. Due to the increase in signal intensity, the resolution might be slightly overestimated under such conditions as the slope might be shifted towards higher voltages. The resolution obtained for TPPAI and THAB monomer is affected by the appearance of a second slope. Sigmoidal fits were adopted to not include this artifact which might have led to a slight underestimation of the resolution. Otherwise, an AMC mobility spectrum of a polydisperse ion distribution will most likely be a combination of several sigmoidal slopes making such an approach necessary.

As a result, the AMC follows the general features of classical DMAs where increases in sheath flow increase the sizing resolution

but decrease the transmission efficiency. The sizing resolution can reach values of 6 to 7 in the presented design. In contrast to the transmission results, no clear size dependence of the sizing resolution is obvious. We found a square root dependence on the ratio of sheath to aerosol flow that contrasts with the linear relationship for classic DMAs (Flagan, 1998; Knutson and Whitby, 1975). To give the reader the possibility to classify the performance of the AMC compared to other instruments, we give an overview in Table 1.

Tammet, 2015, recommended to realize a spacing L of 2 times the tube radius R between the two classifying electrodes i.e., L≥2R, to ensure long parallel electric field lines in the classification region. In Cai and Jiang, 2019, also smaller relationships were studied (R ≥ L), and it was found that this recommendation is not necessary to ensure sufficiently parallel field lines. We also tested insulating spacers of 2 cm width (data not shown), instead of the 0.4 cm that we used otherwise throughout this publication. But apart from the need of higher voltages to cover the size range for the tested cluster ions, it did not result in improved resolution.

With the 0.4 cm spacing, we estimate that, in principle, a range of up to 10 Vs/cm² in terms of inverse mobility or roughly 6.5 nm of mobility equivalent diameter could be classified regarding the breakdown voltage at ambient pressure.

### 3.3. Comparison with a simplified numerical model

Cai and Jiang recently presented a simplified numerical model to describe the transmission efficiency of charged particles for the case of a parabolic Hagen-Poiseuille and a plug flow profile through adverse electric fields which are similar to the one encountered

in the AMC (Cai and Jiang, 2019). They developed their model for the scenario of an electrical mobility filter made of EDM tube, whereas we are dealing with an electric field generated from a ring electrode configuration.

In contrast to the EMF, we utilized a core sampling in the AMC. In the model, the relative transmission is typically determined counting all ions over the whole cross-section at the tube outlet. To incorporate the core sampling into the model we reduced the outlet cross-section to about ¼ matching the instrument geometry. In the model, the size of the core sampling cross-section is of

relevance for the instruments' resolving power with lower size corresponding to higher sizing resolution. Regarding the instrument design, this option is limited by practical considerations regarding geometry and selected flow rates.

In Fig. 8, we show the results of simulated voltage scans with the EMF model at different flow rates for the dimensions of the AMC for a parabolic Hagen-Poiseuille and a plug flow scenario in comparison to experimental data for the THAB dimer at 1.78 nm for different flow rates. There is in general a good agreement between model and experiment even though there are differences

in the electric fields in the EMF and the AMC. Further, the model was designed for much lower flow rates, still it seems to offer reasonable results for the applied flow conditions. Comparing the plug and the parabolic flow profiles of the model, these show distinct differences in the classifying slopes. The experimental data agrees more with the parabolic flow profile. With increasing flow, the lower part of the slope of the experimental data gets steeper compared to the parabolic scenario. This indicates that for the set aerosol flow, the increase in sheath flow allows the flow profile in the AMC to evolve closer into the desired plug flow. In

the model, the expected sizing resolution is higher for an ideal plug flow. It can also be seen that the sizing resolution approaches a limit value for the modelled Hagen-Poiseuille and plug flow scenario following to the square root dependency. The experimental data indicate however that the limit is not yet approached as the resolution increases further for a $Q_{ae}$ of 12 L/min. In Fig. 7, for





$Q_{ae}$ of 5 L/min the resolution approaches the limit. In comparison with the CFD simulations, where certain combinations of aerosol and sheath flow yield approximations of the plug flow profile in the relevant cross-section of the classification region, the model
of Cai and Jiang indicates larger discrepancies to the desired plug flow.

### 3.4. Mass-mobility measurements

Mass-mobility measurements of atmospheric ions are presented in Fig. 9. The mass spectrum is shown as an average over the whole measurement cycle in the right panel. The mass-mobility scan is illustrated as a heat map in the centre. Selected traces are presented in the lower panel.

The AMC was set to flow conditions yielding a transmission through the AMC of about 60 % with grounded electrodes and a sizing resolution of 6.2. During the measurement mainly medium sized ions in the mass range of m/z 200 to 450 were observed. We indicate with the green lines in Fig. 9 where the half-pass mobility for each voltage step can be expected according to mass-mobility relationships though we note that Maißer et al., 2015, showed that no universal mass-mobility relationship exists (Steiner et al., 2014; Mäkelä et al., 1996; Maißer et al., 2015). The green lines are calculated from the half-pass voltage of the three traces
covering the m/z ranges 200-300, 300-400 and 400-500 using the mass-mobility relationships in Steiner et al., 2014, and Mäkelä et al., 1996. The mass-mobility scan shows that the major fraction of ions, especially in the m/z range >200, is classified according to the half-pass mobility following the AMC voltage.

The traces of the two m/z ranges 18-100 and 100-200 did not fit with the expected half-pass mobility obtained from the other m/z ranges, even with the peaks at m/z 102.128 and m/z 186.222 being excluded from the 100-200 m/z range. Regarding the applied
ion transfer tuning we explain the different slopes of both lower m/z ranges from the mass-mobility scan with the AMC-ioniAPi-TOF compared to the higher m/z ranges with the detection of fragments of cluster ions of lower mobility. Cluster ions that decompose during ion transfer dissociate into smaller molecular units whereof one part keeps the charge and is detected with the neutral one not being detected.

We marked the half-pass voltage for two interesting peaks in the heat map, namely m/z 102.128 and m/z 186.222. These belong
to the highest peaks in this lower mass range and their half-pass voltage does not match the value predicted from their mass. The trace of m/z 102.128 has a similar electrical mobility as the m/z range 300-400 suggesting that it might be a fragment of a larger cluster coming from this m/z range. We assume that m/z 102.128 is most likely an C6-amine with the sum formula $C_6H_{15}N \cdot H^+$. C6-amines were reported in different locations like in an urban area, Shanghai (China), as well as at a rural site in Germany with only the latter measurement showing a diurnal cycle which was attributed to re-evaporation from particles (Kürten et al., 2016;
Yao et al., 2016).

The mobility of m/z 186.222 is comparable to the mobility of the m/z range 400 to 500 suggesting it is also a fragment of a decomposed cluster of higher mass as this ion would otherwise have a higher half-pass voltage. We speculate that the peak could be e.g., $C_{12}H_{27}N \cdot H^+$, potentially an amine like tributylamine (Ge et al., 2011).

We also observed that above an AMC voltage of 600 V ions below m/z 100 decreased with a much lower slope even for the
maximum voltage applied during the AMC scan despite expecting them to decrease to zero much faster regarding their high mobility. We suggest that the slope of small ions, especially the hydronium water cluster ions, correlate with larger ions because the slope still decreases at low rates. These larger ions were likely lower in abundance but could still be able to pass even the highest potential barrier applied. However, they could not be detected because of the set limit of the mass range or due to low transmission. From these larger ions and others, the small ions could be the result of cluster decomposition, or of any endothermic
secondary ion chemistry promoted by the RF amplitude applied to the first hexapole (Kambara, Mitsui, and Kanomata 1980).





## 4. Conclusions

We introduced a new type of axial ion mobility classifier, the AMC, with the aim to apply the mobility classification technique to the measurement of atmospheric ions in tandem with mass spectrometry. Compared to the recently presented HP-EMF by Bezantakos et al, 2015, and Surawski et al., 2017, we enhanced the performance characteristics for the sub 3 nm range due to the

successful incorporation of a sheath flow. The influence of flow conditions in the device on transmission efficiency and sizing resolution was investigated. The results demonstrated a broad range of usable flow conditions. Improvements of the inlet geometry will certainly increase the potential resolving power for plug flow conditions as suggested by the model of Cai and Jiang. A sizing resolution of up to 7 was found which is comparable with other state-of-the-art DMAs that are designed for the use of low to medium flow rates in similar size ranges. In comparison to the discussed DMAs, we determined a significantly higher transmission

efficiency under optimum settings of up to 70 %, or 35 % at the half-pass mobility. We also presented first exemplary results from measurements of atmospheric ions with the combined approach of mobility (AMC) and mass spectrometric (ioniAPi-TOF) analysis and could reveal the detection of fragments of cluster ions.

*Data availability.* Data related to this article are available on request from the corresponding authors.


*Author contributions.* GS and ML designed the AMC. ML, GS, LF and AH designed the experiments. SB, JR, and ML performed the experiments with the tandem UDMA-UDMA setup, ML the tandem UDMA-AMC experiments. LF and ML carried out the simulations with OpenFOAM. ML made the simulations with the particle model. ML analysed all the data. ML drafted the manuscript and all authors contributed to the final manuscript development.


*Acknowledgements.* We acknowledge Florian Zweiker and Simon Johann Mayregger of our institute's machine workshop for manufacturing the AMC parts.

*Funding.* This work is funded by the Austrian Science Fund, FWF (project no. P27295-N20), the Tiroler Wissenschaftsfonds

(nanoTOF-ICE), the University of Innsbruck promotion grant for young researchers.

*Competing interests.* The authors declare no competing interests.

**Tables and Figures**


| Instrument | Sizing resolution | Transmission efficiency ($N_2/N_1$) |
|---|---|---|
| TSI nanoDMA 3085 (3086) | 3.9 (+ 10-20%) | 0.07 |
| Grimm S-DMA | 5.41 | 0.03 |
| mini cyDMA | 5.7 | 0.12 |
| half-mini DMA | 6.83 | 0.12 |
| AMC | 6.5 | 0.24 |

Table 1. Comparison of sizing resolution and transmission efficiency of the AMC with cylindrical DMAs that are optimized for low sheath flows. The values of the instruments were taken from Fig. 8 in (Cai et al., 2018). All values were obtained for classification of THAB monomer with 1.48 nm at the typical 1/10 ratio of aerosol to sheath flow for DMAs and AMC and may





not represent the optimum values for each instrument. The values for the transmission efficiency of the AMC correspond to the half-pass mobility and are twice as high without applied AMC voltage.

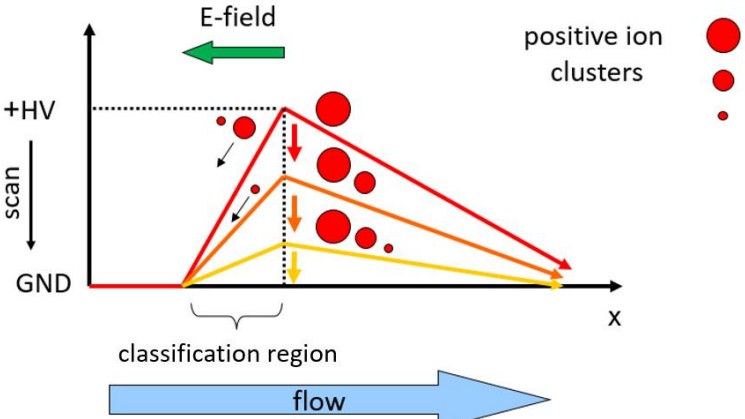

Figure 1. Principle of the axial ion mobility classifier (AMC). An electric field is opposed to the flow direction. The strength of
the electric field decreases from red to yellow. Scanning of the voltage allows to determine the half-pass mobility of a specific ion. The half-pass mobility is found at the voltage where one half of the number concentration of this ion is able to pass the potential barrier.

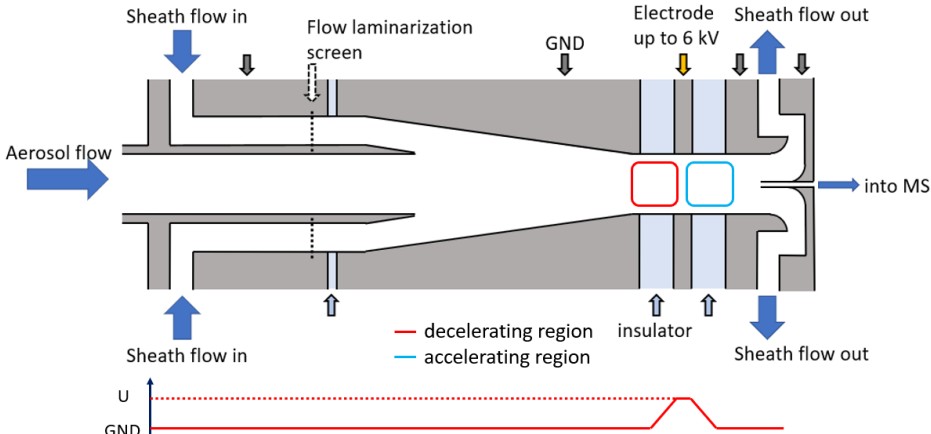

Figure 2. Cross sectional view of the axial ion mobility classifier (AMC). Aerosol flow consisting of ions of various sizes enters
the inlet from the left side. A recirculating sheath flow is introduced radially and laminarized via a nylon screen. Sheath flow and aerosol flow are mixed within a cone. A voltage can be applied and varied on a lens that is insulated with Teflon spacers for mobility classification. Ions exit through a core sampling and are transferred to a mass spectrometer (MS) for detection. Sheath flow and excess aerosol flow are pumped after the core sampling.






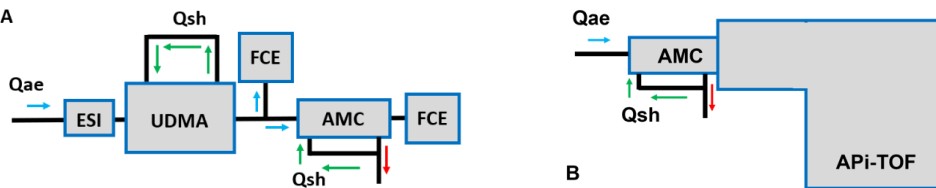

Figure 3. Experimental setups. A) Transmission characterization of the AMC using a Cluster Calibration Unit consisting of bipolar Electrospray, UDMA, a y-shaped flow splitter and two Faraday Cup Electrometers (FCE). Aerosol flow, Qae, indicated with light blue arrows. Sheath flow, Qsh, in green arrows and exhaust flow in red. B) Setup for ambient measurements in Innsbruck, Austria, with the AMC-ioniAPi-TOF.


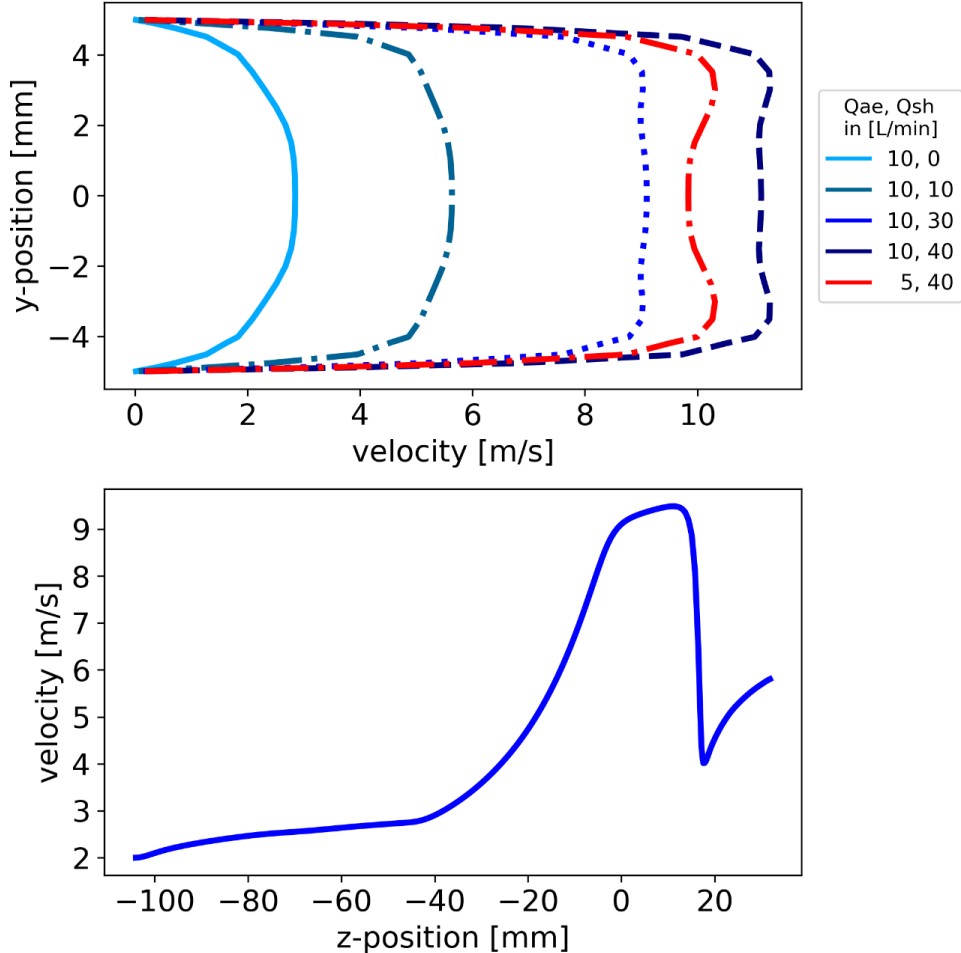

Figure 4. In the upper panel, radial velocity profiles in the AMC inlet at the presumed z-position, 0 mm, where the size segregation mainly takes place (marked with a white arrow in Fig. S7). The radial velocity profiles are shown for four flow settings. From the five blower settings that we used in the experiments to generate the sheath flow, we simulated conditions for the first, the second,






and the fifth's blower setting corresponding approximately to sheath flows of 0 L/min, 55 L/min and of 105 L/min, respectively. In the lower panel, the velocity profile in z-direction is shown for the combination of 10 L/min sample and 55 L/min sheath flow.

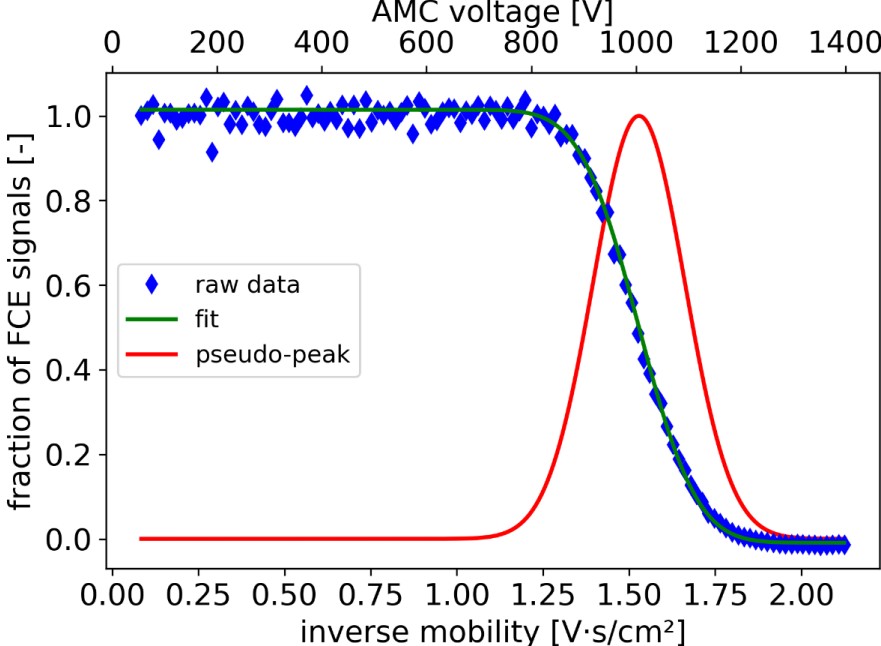

Figure 5. Example AMC scan of THAB dimer showing the relative transmission efficiency i.e., the fraction of FCE signal after the AMC (FCE2) divided by the FCE in front of the AMC (FCE1), as a function of the scanning voltage. Raw data presented in blue follows a sigmoidal relationship displayed as green line. The derivative of the fitted sigmoidal function multiplied by -1 yields a pseudo-peak shown in red.




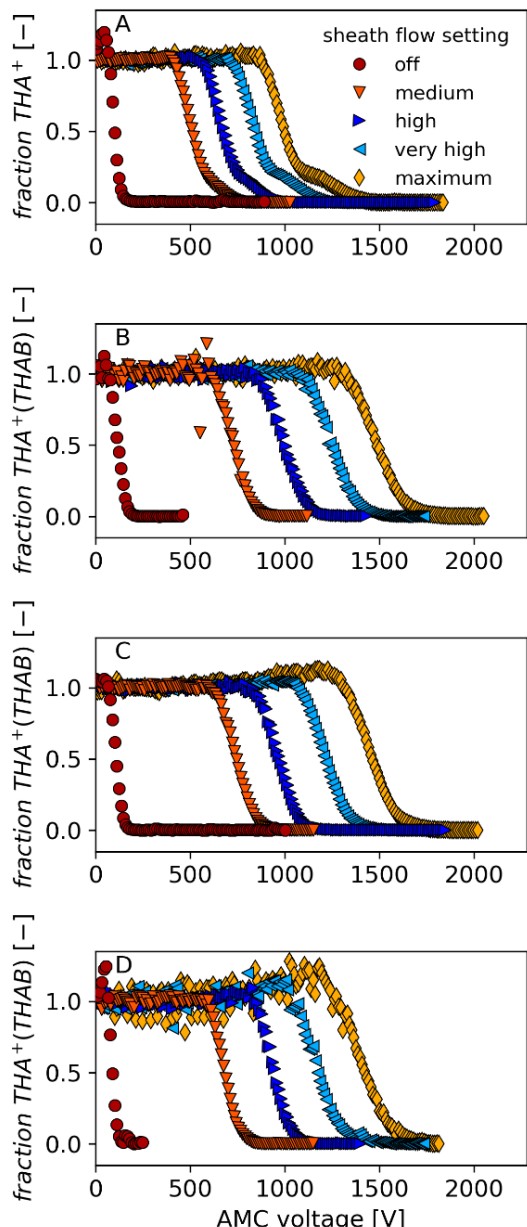

Figure 6. Demonstration of the AMC scanning procedure for two selected ions at three aerosol flow, $Q_{ae}$, and five different sheath flow, $Q_{sh}$, settings. Panel A) shows the THAB monomer (A+) at a $Q_{ae}$ of 12.4 L/min, B) the THAB dimer (A+(AB)$_1$) at a $Q_{ae}$ of 12.4 L/min, C) the THAB dimer at a $Q_{ae}$ of 9.6 L/min and D) the THAB dimer at a $Q_{ae}$ of 5.0 L/min. Data of all other experiments can be found in the supplement. The sheath flow settings correspond to flow rates of approximately: "off": 0 L/min, "medium": 50 L/min, "high": 70 L/min, "very high": 85 L/min, "maximum": 105 L/min.





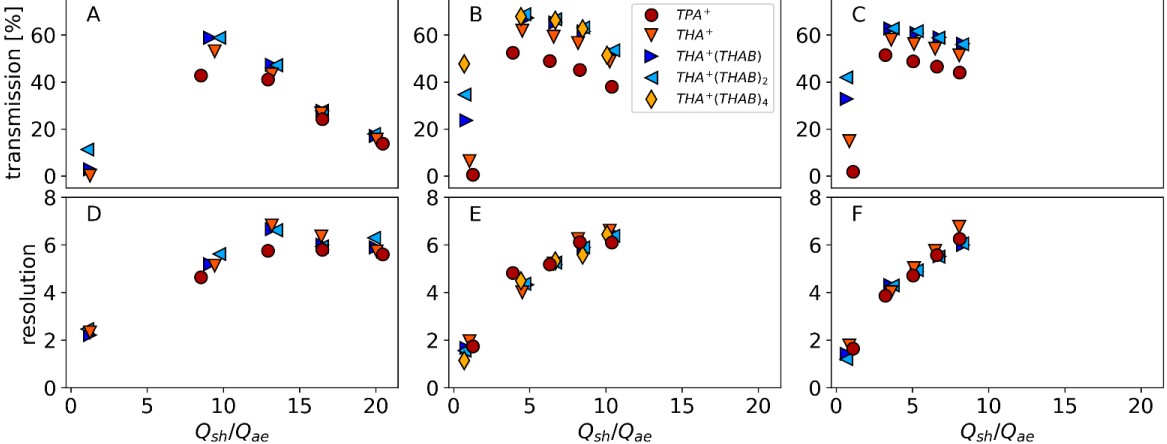

Figure 7. The transmission (upper panel) and resolution (lower panel) are characterized for three aerosol flow rates: A and D) $Q_{ae}$ = 5.0 L/min, B and E) $Q_{ae}$ = 9.6 L/min, C and F) $Q_{ae}$ = 12.4 L/min Experiments were done for selected ions in the range of inverse mobilities from 0.619 up to 2.79 Vs/cm² (TPPAI monomer: 0.619 Vs/cm²; THAB monomer: 1.03 Vs/cm², dimer: 1.529 Vs/cm², trimer: 1.893 Vs/cm² and pentamer: 2.79 Vs/cm²).

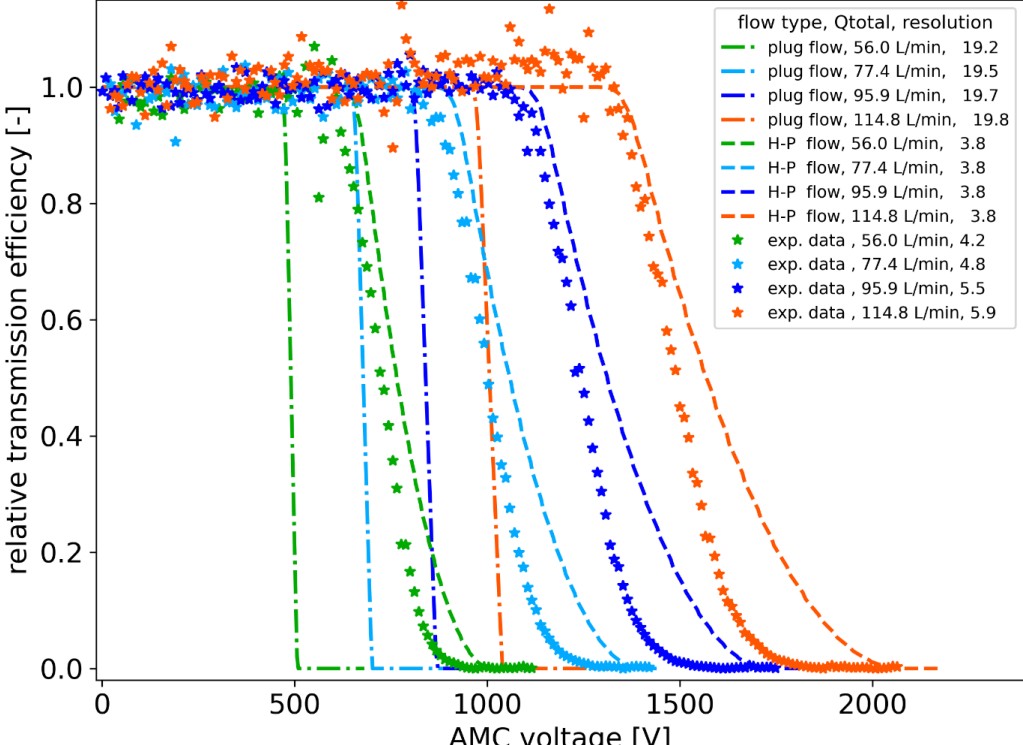

Figure 8. Comparison of modelled EMF classification curves for plug and laminar (Hagen-Poiseuille, H-P) flow profiles with experimental data (exp. data) obtained with the AMC. The total flow rates and the respective sizing resolutions are noted in the legend. The experimental data is based on the flow settings with a $Q_{ae}$ of 12 L/min.



**Figure 9.** Exemplary mass-mobility scan of ambient ions from 10 am to 4 pm on 12.03.2020. The AMC voltage integration step width is 26 V. The green lines represent the extrapolated half-pass voltage (mobility) for all m/z estimated from the average half-pass voltage for the three mass ranges 200 to 300, 300 to 400 and 400 to 500. In the right panel, a mass spectrum integrated over all AMC voltages is shown. The lower panel shows selected m/z traces during the AMC scan. The line in light blue indicates the half-pass mobility voltage of m/z 102.128, and the dark blue line to m/z 186.222.

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
