# Peer review of "A high transmission axial ion mobility classifier for mass-mobility measurements of atmospheric ions"

_Atmospheric Measurement Techniques, 2022_

## Referee Comment (RC2)

Review of "A high transmission axial ion mobility classifier for mass-mobility measurements of atmospheric ions" by Leiminger et al. (amt-2022-29)

**General comments:**

This paper describes an electrical mobility classifier for mass-mobility measurements of atmospheric ions. The authors developed an axial ion mobility classifier (AMC) coupled with sheath flows and achieved high transmission efficiencies to segregate natural ions at comparable sizing resolution to previous devices. The mass-mobility measurement of atmospheric ions was demonstrated by AMC-ioniAPi-TOF. By combining mass spectrometry with mobility classifier technique, fragment ions from cluster ions could be distinguished. I think that this work was well-conducted and that the paper is generally well-written. I recommend this paper to be published in Atmospheric Measurement Techniques after the authors' consideration of my minor comments detailed below.

**Specific comments:**

(1) Page 1, Line 22 and Page 11, Line 379: In Abstract and Conclusions, the authors used "sub 3 nm", but in the text it is just written that cluster ions of 1 to 4 nm were formed (Page 4, Line 137). Did the authors explain that the improvement of the size resolution was achieved only in the sub 3 nm range in the text? The size of THAB monomer is written to be 1.48 nm (Page 8, Line 287). I think that it is better to show the size of THAB dimer, THAB trimer, THAB pentamer, and TPPAI monomer if possible.

(2) Page 5, Lines 152−153: It is written that the blower was set to voltages from 0 to 5 V in 1 V step. This might mean that there are 6 different conditions. But in the previous sentence, it is written that the calibration was carried out in 5 different blower settings. Are these two sentences consistent each other?

(3) Page 6, Line 187: What is CFD the abbreviation of?

(4) Page 6, Lines 210−211: I couldn't understand the sentence "At a medium sheath flow setting of 55 L/min the impact of aerosol flow rate becomes clear comparing the flow profiles for aerosol flows of 5 and 10 L/min". What became clear?

What is the value of Qsh at the medium sheath flow setting of 55 L/min? I supposed that it was 30 L/min of Qsh because the profile shown in the lower panel of Figure 4 was obtained with 10 L/min sample and 55 L/min sheath flow. But I might be wrong from the caption of Figure 6. If the value of Qsh at the medium sheath flow setting of 55 L/min is 10 L/min, is the lower panel of Figure 4 consist with the results of the upper panel of Figure 4? I was confused.

(5) Page 7, Line 242: I cannot judge if it has a higher resolution at higher sheath flows or not. Did the authors calculate the resolution according to eqn. (3)? Or is the resolution derived from eqn. (3) determined mainly from $z_{1/2}$?

(6) Page 7, Lines 250−251: I felt that the phrase "the previous discussion" here is ambiguous.

(7) Page 8, Line 267: Define "U". Probably it is the same as "AMC voltage" in Figs. 5 and 6.

(8) Page 8, Line 286: In Figs. 5 and 6, the values of the y-axis at AMC voltage ≈ 0 are almost 1 because they are "relative" values. If the authors plot these figures with the absolute transmission efficiency in the y-axis, are values at zero AMC voltage close to 0.7? If so, this information should be better to be mentioned when the authors explain Figure 5.

(9) Page 8, Line 290: I couldn't understand how the authors derived the value of "48.6 %". In which figure does it show?

(10) Page 8, Line 293: In the text (also in Line 273), the ratio $Q_{total}/Q_{ac}$ is mentioned. However, the x-axis of Figure 7 is $Q_{sh}/Q_{ac}$. Is it ok?

(11) Page 10, Line 341: The results shown in this section are very interesting and valuable. I would like to know the background level of this system.

**Technical comments:**

(1) Page 1, Line 21: I could not find "Surawaski et al., 2017" in References.

(2) Page 4, Line 129: I could not find "Fu et al., 2019" in References.

(3) Page 5, Line 155: I could not find "Leiminger et al., 2019" in References.

(4) Page 7, Line 257: Fig. 5 → Fig. 6

(5) Page 8, Line 281: Surawaski et al. → Surawski et al., 2017. Is it correct?

(6) Page 9, Line 310: I could not find "Tammet, 2015" in References.

(7) Page 10, Line 342: It is better to mention Fig. 3B here.

(8) Page 10, Lines 362 and 368: "+" should be a superscript.

(9) Page 19, Line 549: "Hideki Kambara" → "Kambara H." and move it to the correct position.

(10) Page S-2 of the Supplementary material, Line 17: I think that the explanation "the sheath flow is set to zero in the first row, and increases from the second to the last row" is wrong. The results of different ions are shown in the different rows.

(11) Page S-2 of the Supplementary material, Line 24: A Germany sentence is inserted.

(12) Page S-3 of the Supplementary material, Line 36: Show Brilke et al. (2019) in Reference of the Supplementary material.

---

## Author Comment (AC1)

**Author's Comments on* "A high transmission axial ion mobility classifier for mass-mobility measurements of atmospheric ions" *by Leiminger et al. – amt-2022-29**

We thank Reviewer #1 for the comments and suggestions, which helped to improve the manuscript. In the following, we present our reply. The reviewer's comments (RC) are marked in *italic*, our replies (AC: author comment) follow in roman and the improved manuscript text is shown in blue.

**Response to Reviewer #1:**
(RC 1-x: reviewer #1 - comment x)

*RC 1-1:* *L132: 'The sheath flow is recirculated via a brushless blower (model code: 465.3.265-841) from Domel, Slovenia, at experimentally determined flow rates of approximately 50 to 115 L/min.' I was confused by this expression, as the sheath flow rate is obviously an important parameter for the AMC. Detailed descriptions about the measurement and control of the sheath flow would be needed.*

AC 1-1: We thank the reviewer for this comment. We agree that the reader would benefit from a more detailed description of the sheath flow control. We changed the manuscript text accordingly.

Line 132: The sheath flow is recirculated via a brushless blower (model code: 465.3.265-841) from Domel, Slovenia. The rotation speed of this device can be controlled with a pulse width modulated (PWM) signal. We used an Arduino uno board (Elegoo UNO R3 Controller Board) to set the speed of the blower. The Arduino offers an output voltage of 0 to 5 V. These can be realized by PWM in 256 discrete steps. The blower would allow up to 10 V as input. The maximum output voltage of 5 V from the Arduino board turned out to be sufficient for a full characterization of the instrument. We figured out that the blower did not start for voltages below about 2 V. Therefore, we selected voltage steps of 0, 2, 3, 4 and 5 V allowing to run the blower at five different speeds to generate the sheath flow of the AMC. Taking the known electrical mobility Z of the THAB monomer as the half-pass mobility $Z_{1/2}$ and the corresponding AMC voltage determined from the sigmoid fit, we estimated sheath flow rates of roughly 50 to 105 L/min with Equ. 5, see section 3.2.3.

Line 165: The blower was set to voltages 0, 2, 3, 4 and 5 V.

Line 278: Here, r is the tube radius, $Q_{total}$ the **total** flow rate, $\Delta\phi$ the electric potential difference **obtained from the sigmoid fit of the AMC scan corresponding to $Z_{1/2}$**.

*RC 1-2:* *Figure S1. It seems that the values for y-axis (e.g., fraction of THAB 3er (-)) are occasionally smaller than zero. I am wondering how the data were measured/analysed.*

AC 1-2: In the supplement, we only show pre-processed data. Here, the offset of the electrometer is not yet accounted for. The final data processing involved the following steps: A) electrometer offset correction, B) conversion from ion current in units of fA (femto Ampere) corresponding to particle number concentration in units of $cm^{-3}$, C) correction for the different flow rates passing the detectors of the electrometers 1 and 2, D) correction factor accounting for systematic differences in ion current measurement from the electrometer position. We added the following sentence in the caption of Fig. S1. The figure shows pre-processed data prior electrometer offset correction.

RC 1-3: L257: 'Fig. 5 panels C and D.' - Figure 6?

AC 1-3: Correct, we wanted to address Fig. 6. We corrected the expression in the main text. Line 257: Fig. 6 panels C and D

RC 1-4: L268: 'We attributed the reason for this initial increase in transmission to the combined effect of sub-isokinetic sampling in the region of the core sampling and elevated electric fields that might reach into the core sampling as is exemplarily illustrated in Fig. S5.'

AC 1-4: We explain now in more detail the observed effect. We explain the reason for the observed initial increase in transmission with the following. In Fig. S5 the electric field lines inside the AMC device are illustrated. Some of the field lines reach the entrance of the core sampling. Considering the flow profile in z-direction, see the lower panel of Fig. 4, there is a sharp drop in flow velocity in the tube centre at the entrance of the core sampling. In front of the core sampling the flow streamlines split into a small sample flow being drawn into the core sampling that reaches the detector and a much larger flow being pulled to the blower and the exhaust. The larger the difference between aerosol- and sheath flow the smaller is the area of the aerosol streamlines. With a smaller area of the aerosol flow at high sheath flows, the way for the discarded aerosol flow around the core sampling is longer. A combination of a lower ion drift velocity at this location and elevated electric fields pointing into the core sampling might force additional ions to enter the core sampling instead of following the streamlines.

RC 1-5: L293: 'The results of the sizing resolution in relation to the ratio Q total /Q ae are shown in the lower panels of Fig. 7.' How was the resolution defined? It needs to be explicitly explained.

AC 1-5: We would like to point to section 3.2.1. where we describe how we define the resolution. In addition, we will update Fig. 5 with two additional lines to indicate the parameters used for determining the resolution to improve the interpretability. We will add the following to the text and the figures caption.

Line 226: The half-pass mobility at the maximum of the pseudo-peak corresponds to the electrical mobility of the ion of interest, **as indicated in Fig. 5**.

Line 442: The black line indicates the half-pass mobility, $Z_{1/2}$, and the dashed black line the FWHM, $\Delta Z$.

*RC 1-6:* *Figure 8, Figure caption/legend does not tell what the dashed lines are. They need to be updated.*

AC 1-6: We agree. We will update the line styles of Fig. 8 accordingly.

*RC 1-7:* *Figure 9, x-axis of the upper panel is hidden by the lower panel. It needs to be modified.*

AC 1-7: In this figure, the x-axis of the lower and the upper panel is the same and therefore we removed the duplicate x-axis of the upper panel to allow reduction of the overall size of this large figure. The same was done with the y-axis of the upper and the right panel. We added a comment in the figure caption raising attention to the reader about this fact.

In the central panel of the figure, an exemplary mass-mobility scan of ambient ions, averaged from 10 am to 4 pm on 12.03.2020, is shown. The lower panel shows selected m/z traces during the AMC scan. **The central and the lower panel share the same horizontal axis.** In the right panel, a mass spectrum integrated over all AMC voltages is shown. The two magenta-coloured lines represent the extrapolated half-pass voltage (mobility) for all m/z estimated from the average half-pass voltage for the three mass ranges 200 to 300, 300 to 400 and 400 to 500. The line in light blue indicates the half-pass mobility voltage of m/z 102.128, and the dark blue line to m/z 186.222. The AMC voltage integration step width is 26 V.

---

## Author Comment (AC2)

*Author's Comments on* "A high transmission axial ion mobility classifier for mass-mobility measurements of atmospheric ions" *by Leiminger et al. – amt-2022-29*

We thank Reviewer #2 for the comments and suggestions, which helped to improve the manuscript. In the following, we present our reply. The reviewer's comments (RC) are marked in *italic*, our replies (AC: author comment) follow in roman and the improved manuscript text is shown in blue.

**Response to Reviewer #2:**

**Specific comments**:
(RC 2-x: reviewer #2 - comment x)

*RC 2-1:        Page 1, Line 22 and Page 11, Line 379: In Abstract and Conclusions, the authors used "sub 3 nm", but in the text it is just written that cluster ions of 1 to 4 nm were formed (Page 4, Line 137). Did the authors explain that the improvement of the size resolution was achieved only in the sub 3 nm range in the text? The size of THAB monomer is written to be 1.48 nm (Page 8, Line 287). I think that it is better to show the size of THAB dimer, THAB trimer, THAB pentamer, and TPPAI monomer if possible.*

AC *2*-1:        We thank reviewer #2 for his comment. We did not intend to confuse any reader and hope to clarify this remark with the following answers. 1) We used the expression "sub 3 nm" to provide the reader with details how to relate this work with others. 2) We did not observe any obvious relation between size and resolution. 3) In Line 137 we wrote: "*enabling the precise formation of cluster ions of 1 to 4 nm as previously shown in (Brilke et al., 2020)*". With this statement, we only wanted to make the reader aware that the bipolar electrospray that we used, has recently been demonstrated to generate ions in the 1 to 4 nm range. To make the point clearer, we updated the following sentence and added the details about the size of the used standard mobility ions as suggested by the reviewer.

Line 135: Brilke et al., 2020, recently demonstrated that the benefit of using a bipolar ESI is that charged droplets of opposite polarity can be generated leading to a favourable reduction of multiple charged ions enabling the precise formation of cluster ions of 1 to 4 nm (Brilke et al., 2020).

Line 146: With the UDMA, we selected well-defined cluster ions. These were the TPPAI monomer as well as the THAB monomer, dimer, trimer and pentamer with mobility diameters of 1.16 nm, 1.48 nm, 1.78 nm, 1.97 nm, and 2.26 nm, respectively.

*RC 2-2:        Page 5, Lines 152-153: It is written that the blower was set to voltages from 0 to 5 V in 1 V step. This might mean that there are 6 different conditions. But in the previous sentence, it is written*

*that the calibration was carried out in 5 different blower settings. Are these two sentences consistent each other?*

AC 2-2: We thank the reviewer #2 for pointing to this inconsistency. While answering to reviewer #1's question *RC 1-1* in AC 1-1, we also explained the reason behind the difference between 6 available logical conditions and the 5 blower settings.

*RC 2-3: Page 6, Line 187: What is CFD the abbreviation of?*

AC 2-3: We thank the reviewer for this remark. We added the full title, please see below.

Computational fluid dynamics (CFD) simulations

*RC 2-4: Part1) Page 6, Lines 210-211: I couldn't understand the sentence "At a medium sheath flow setting of 55 L/min the impact of aerosol flow rate becomes clear comparing the flow profiles for aerosol flows of 5 and 10 L/min". What became clear?*

AC 2-4: We agree that the statement can be improved for better readability.

A comparison of the different aerosol flows of 5 and 10 L/min indicates how much the flow profile deviates from the desired plug flow profile for a fixed sheath flow rate.

*RC 2-4: Part 2) What is the value of Qsh at the medium sheath flow setting of 55 L/min? I supposed that it was 30 L/min of Qsh because the profile shown in the lower panel of Figure 4 was obtained with 10 L/min sample and 55 L/min sheath flow. But I might be wrong from the caption of Figure 6. If the value of Qsh at the medium sheath flow setting of 55 L/min is 10 L/min, is the lower panel of Figure 4 consist with the results of the upper panel of Figure 4? I was confused.*

AC 2-4: We understand the confusion about Qsh. We would like to point out that the Qsh values shown in the legend of Figure 4 differ from the one in the discussion about Fig. 6. We now also realized that the values for Qsh are not in line with the values given in the caption of Fig. 6. To improve the consistency of the manuscript, we propose to update Figure 4: we remove the flow profiles of flow rates that are not used in later parts of the manuscript that could lead to confusion (lines with Qsh = 10 and 30 L/min, see in the legend). On the other hand, we would like to add the flow profile of the "maximum" flow rate setting which is discussed in the main text but missing in the figure. The lower panel of Figure 4 remains unchanged.

*RC 2-5: Page 7, Line 242: I cannot judge if it has a higher resolution at higher sheath flows or not. Did the authors calculate the resolution according to eqn. (3)? Or is the resolution derived from eqn. (3) determined mainly from z1/2?*

AC 2-5: The resolution was calculated using equation (3) for every single ion at all tested flow settings. As we used ion mobility standards (Ude and De La Mora, 2005), we defined $Z_{1/2}$ being equal to the mobility of the tested ion e.g., $Z_{1/2} = Z(THA^+) = 0.97$ cm$^2$/Vs. The full width at half maximum of

$Z_{1/2}$, which is $\Delta Z$, was determined using the derivative of a sigmoid fit as described in section 3.2.1. As stated earlier in the reply to reviewer #2's comment, *RC 1-5*, we will update Figure 5 to improve the understanding how we determined the resolution. That the resolution generally is higher at higher sheath flows can be taken by the increase in ratio Qsh/Qae with fixed Qae in Fig. 7 panels D, E and F.

*RC 2-6:*      *Page 7, Lines 250-251: I felt that the phrase "the previous discussion" here is ambiguous.*

AC *2-6:*      We thank the reviewer for his suggestion and removed the phrase.

This effect cannot be ruled out completely, but it seems to be less likely.

*RC 2-7:*      *Page 8, Line 267: Define "U". Probably it is the same as "AMC voltage" in Figs. 5 and 6.*

AC *2-7:*      The reviewer is correct, "U" is the same as "AMC voltage". We defined the expression in the main text.

With U as the voltage applied to the AMC electrode, the corrected ratio of $N_{FCE2(U=0V)}$ to $N_{FCE1}$ yields the transmission efficiency for a single setting.

*RC 2-8:*      *Page 8, Line 286: In Figs. 5 and 6, the values of the y-axis at AMC voltage ≈ 0 are almost 1 because they are "relative" values. If the authors plot these figures with the absolute transmission efficiency in the y-axis, are values at zero AMC voltage close to 0.7? If so, this information should be better to be mentioned when the authors explain Figure 5.*

AC *2-8:*      Yes, if we plot both figures with the absolute transmission efficiency on the y-axis, the values are close to 0.7 for an AMC voltage of zero. We will add the absolute values on a second y-axis on the right in Fig. 5 to give the reader an example and will mention it in the caption.

Line 442: The absolute transmission efficiency is given on the right y-axis.

Line 223: How this affects the absolute transmission can be taken from the right y-axis.

In Fig. 6, we preferentially used the relative transmission for reasons of clarity. We added this statement in the caption of Fig. 6.

For reasons of visibility, we only show the relative transmission, "fraction"; and not the absolute transmission efficiency.

*RC 2-9:*      *Page 8, Line 290: I couldn't understand how the authors derived the value of "48.6 %". In which figure does it show?*

AC *2-9:*      The transmission of 48.6 % can be found in Fig. 7B for the THAB monomer, $THA^+$, at a $Q_{sh}/Q_{ae}$ of about 10. In line 286, we describe in two sentences how we derived the value.

Line 290: In this case, **see Fig. 7B**, the AMC offers a transmission of 48.6 % when no voltage is applied.

*RC 2-10:* *Page 8, Line 293: In the text (also in Line 273), the ratio Qtotal/Qac is mentioned. However, the x-axis of Figure 7 is Qsh/Qac. Is it ok?*

AC 2-10: We thank the reviewer for making us aware of this inconsistency. The ratio of $Q_{sh}/Q_{ae}$ as given in Fig. 7 is correct while it is not correct in the main text. We corrected this accordingly.

In section 3.2.3: $Q_{total}/Q_{ae}$ was replaced by $Q_{sh}/Q_{ae}$ in the main text.

*RC 2-11:* *Page 10, Line 341: The results shown in this section are very interesting and valuable. I would like to know the background level of this system.*

AC 2-11: We appreciate the reviewer's interest. The background level of the used mass spectrometer is about $0.00025$ cm$^{-3}$ for 1 h integration time over a mass range up to 900 amu.

**Technical comments:**

(Tx: technical comment x )

*RC 2-T1:* *Page 1, Line 21: I could not find "Surawaski et al., 2017" in References.*

AC 2-T1: We added "Surawski et al. 2017" to the list.

Surawski, N. C., Bezantakos, S., Barmpounis, K., Dallaston, M. C., Schmidt-Ott, A., and Biskos, G.: A tunable high-pass filter for simple and inexpensive size-segregation of sub-10-nm nanoparticles, Sci. Rep., 7, 1–8, https://doi.org/10.1038/srep45678, 2017.

*RC 2-T2:* *Page 4, Line 129: I could not find "Fu et al., 2019" in References.*

AC 2-T2: We added this reference to the list.

Fu, Y., Xue, M., Cai, R., Kangasluoma, J., and Jiang, J.: Theoretical and experimental analysis of the core sampling method: Reducing diffusional losses in aerosol sampling line, Aerosol Sci. Technol., 53, 793–801, https://doi.org/10.1080/02786826.2019.1608354, 2019.

*RC 2-T3:* *Page 5, Line 155: I could not find "Leiminger et al., 2019" in References.*

AC 2-T3: We added it to the list.

Leiminger, M., Feil, S., Mutschlechner, P., Ylisirniö, A., Gunsch, D., Fischer, L., Jordan, A., Schobesberger, S., Hansel, A., and Steiner, G.: Characterisation of the transfer of cluster ions through an atmospheric pressure interface time-of-flight mass spectrometer with hexapole ion guides, Atmos. Meas. Tech., 12, 5231–5246, https://doi.org/10.5194/amt-12-5231-2019, 2019.

*RC 2-T4:* *Page 7, Line 257: Fig. 5 → Fig. 6*

AC 2-T4: We agree. We corrected it in the main text.

Line 257: Fig. 6 panels C and D.

*RC 2-T5:*        *Page 8, Line 281: Surawaski et al. → Surawski et al., 2017. Is it correct?*

AC T5:        We are sorry, but we cannot find the expression "Suraw**a**ski et al.". It appears to be correct in the original submission.

*RC 2-T6:*        *Page 9, Line 310: I could not find "Tammet, 2015" in References.*

AC *2*-T6:        We added this reference.

Tammet, H.: Passage of charged particles through segmented axial-field tubes, Aerosol Sci. Technol., 49, 220–228, https://doi.org/10.1080/02786826.2015.1018986, 2015.

*RC 2-T7:*        *Page 10, Line 342: It is better to mention Fig. 3B here.*

AC *2*-T7:        We agree. This improves the understanding.

Mass-mobility measurements of atmospheric ions are presented in Fig. 9. In this experiment, we used the setup depicted in Fig. 3B with the AMC coupled to inlet of the ioniAPi-TOF.

*RC 2-T8:*        *Page 10, Lines 362 and 368: "+" should be a superscript.*

AC *2*-T8:        We agree with the reviewer. We corrected the expression in the main text.

Line 362: $C_6H_{15}N\cdot H^+$, Line 368:  $C_{12}H_{27}N\cdot H^+$,

*RC 2-T9:*        *Page 19, Line 549: "Hideki Kambara" → "Kambara H." and move it to the correct position.*

AC *2*-T9:        We thank the reviewer for this detail and corrected it accordingly.

Kambara, H., Mitsui, Y., and Kanomata, I.: Mass spectrometric study of ions produced in oxygen at atmospheric pressure by a collisional dissociation method, Int. J. Mass Spectrom. Ion Phys., 35, 59–72, 1980.

*RC 2-T10:*        *Page S-2 of the Supplementary material, Line 17: I think that the explanation "the sheath flow is set to zero in the first row and increases from the second to the last row" is wrong. The results of different ions are shown in the different rows.*

AC *2*-T10:        The reviewer is correct. We corrected the sentence in the caption.

For each ion, the AMC voltage is scanned for different sheath flow settings (off: 0 L/min, medium: 50 L/min, high: 70 L/min, very high: 85 L/min, maximum: 105 L/min).

*RC 2-T11:*        *Page S-2 of the Supplementary material, Line 24: A Germany sentence is inserted.*

AC *2*-T11:        We corrected the sentence.

In linear scale, the slope in the carried-out AMC scan of THAB monomer resembles the slopes in Figure 6 and Figure S4.

*RC 2-T12:      Page S-3 of the Supplementary material, Line 36: Show Brilke et al. (2019) in Reference of the Supplementary material.*

AC *2*-T12:      As suggested by the reviewer, we added the reference on the last page of the supplementary material.

Brilke, S., Resch, J., Leiminger, M., Steiner, G., Tauber, C., Wlasits, P. J., and Winkler, P. M.: Precision characterization of three ultrafine condensation particle counters using singly charged salt clusters in the 1–4 nm size range generated by a bipolar electrospray source, Aerosol Sci. Technol., 0, 1–14, https://doi.org/10.1080/02786826.2019.1708260, 2020.